# Mutation Mechanism of Leaf Color in Plants: A Review

**Ming-Hui Zhao, Xiang Li, Xin-Xin Zhang, Heng Zhang and Xi-Yang Zhao ***

State Key Laboratory of Tree Genetics and Breeding, School of Forestry, Northeast Forestry University,
Harbin 150040, China; zhaominghui66@163.com (M.-H.Z.); lx2016bjfu@163.com (X.L.);
zhangxinxin@nefu.edu.cn (X.-X.Z.); zhangheng815@nefu.edu.cn (H.Z.)
* Correspondence: zhaoxyphd@163.com; Tel.: +86-0451-8219-2225

**Abstract:** Color mutation is a common, easily identifiable phenomenon in higher plants. Color mutations usually affect the photosynthetic efficiency of plants, resulting in poor growth and economic losses. Therefore, leaf color mutants have been unwittingly eliminated in recent years. Recently, however, with the development of society, the application of leaf color mutants has become increasingly widespread. Leaf color mutants are ideal materials for studying pigment metabolism, chloroplast development and differentiation, photosynthesis and other pathways that could also provide important information for improving varietal selection. In this review, we summarize the research on leaf color mutants, such as the functions and mechanisms of leaf color mutant-related genes, which affect chlorophyll synthesis, chlorophyll degradation, chloroplast development and anthocyanin metabolism. We also summarize two common methods for mapping and cloning related leaf color mutation genes using Map-based cloning and RNA-seq, and we discuss the existing problems and propose future research directions for leaf color mutants, which provide a reference for the study and application of leaf color mutants in the future.

**Keywords:** color mutation; pigment metabolism; chlorophyll; anthocyanin; mutation mechanism; RNA-seq

## 1. Introduction

As part of photosynthesis, leaves play an important role in the growth and development of plants. Leaf color mutation is a high frequency character variation that is easy to recognize and ubiquitous in various higher plants [1]. Leaf color mutations usually affect the plant's photosynthetic efficiency, resulting in stunted growth and even death. Therefore, leaf color mutations have been considered harmful mutations with no practical value by researchers in the past [2,3]. Since Granick used the Chlorella vulgaris green mutant *W5* to validate that protoporphyrin III was a precursor to chlorophyll synthesis in 1948, research related to leaf color mutants has gradually gained attention, especially research related to chlorophyll synthesis [4,5]. Physical and chemical mutagenesis and tissue culture are usually used to induce leaf color mutations and obtain leaf color mutants [6]. In general, the mutant genes of leaf color mutants can directly or indirectly affect the pigment (such as chlorophyll and anthocyanin) synthesis, degradation, content and proportion, which can block photosynthesis and lead to abnormal leaf color. Leaf color mutations are generally expressed at the seedling stage and can be divided into eight types: albino, greenish-white, white emerald, light green, greenish-yellow, etiolation, yellow-green, and striped [1]. In addition, leaf color mutants can also be divided into four types: total chlorophyll increased type, total chlorophyll deficient type, chlorophyll a deficient type, and chlorophyll b deficient type [7]. As an especially ideal material, leaf color mutants play an important role in the research of photosynthetic mechanisms, the chlorophyll biosynthesis

pathway, chloroplast development and genetic control mechanisms [8,9]. Leaf color mutants have been obtained in *Zea mays* [10,11], *Populus* L. [12,13], *Nicotiana tabacum* [14–16], *Arabidopsis thaliana* [17–20], *Oryza sativa* [21–24], *Rosa multiflora* [25,26], and *Cucumis melo* [27], resulting in many studies being conducted in these species.

Genetic changes in plant cells are usually the cause of leaf color mutations, although the underlying mechanism is more complicated. There are more than 700 sites involved in leaf color mutations in higher plants, all of which are involved in the development, metabolism or signal transduction of leaf color formation. We can therefore analyze and identify gene functions and understand gene interactions by using these mutants [28]. This paper mainly reviews studies related to leaf color mutants, describing some genes that control or affect pigment metabolism, chloroplast development and differentiation, and photosynthesis. We summarize two methods for mapping and identifying leaf color mutation genes, and preliminarily elucidate the formation mechanism of leaf color mutations. We also present the achievements and challenges inherent to the study of leaf color mutants and future research directions to provide a reference for the future study and application of leaf color mutants.

## 2. Genetic Model of Plant Leaf Color Mutants

Many studies have proved that the genetic modes of plant leaf color mutants are mainly divided into three types: nuclear heredity, cytoplasmic heredity and nuclear-plasmid gene interaction heredity, among which nuclear heredity is the most important genetic mode of leaf color. Studies of leaf color mutants mainly focus on recessive mutants controlled by nuclear genes. Such mutants follow Mendelian inheritance laws, including single-gene inheritance and multigene inheritance, among which the single gene recessive inheritance mode results in the majority of leaf color mutants and leaf color inheritance. A series of white leaf color mutants *virescen1*, *virescen3*, and yellow-green leaf *YSA* mutants were found in rice, all of which were proved to be controlled by a pair of recessive nuclear genes [28–30]. In addition, leaf color mutations controlled by a single recessive gene were also found in such species as *Capsicum annuum*, *Cucumis sativus*, and *C. melo* [27,31,32]. Moreover, it was found that the Sesamum indicum yellow-green mutation was controlled by an incompletely dominant nuclear gene, *Siyl-1* [33]. Conversely, few studies and reports exist related to leaf color mutations caused by mutations of the cytoplasmic genes and nuclear cytoplasmic gene interactions in such species as *A. thaliana*, *N. tabacum*, and *Lycopersicon esculentum* [34–36]. These mutations may be related to the fact that plant cells contain multiple organelles (e.g., chloroplasts and mitochondria) with their own DNA molecules.

## 3. Molecular Mechanisms of Plant Leaf Color Mutations

The molecular mechanisms of leaf color mutations are complex. The mutated genes can directly or indirectly interfere with pigment synthesis and stability, resulting in various sources of leaf color mutations. In this paper, we summarized the molecular mechanisms underlying the formation of leaf color mutants through the following aspects.

### 3.1. Abnormal Chlorophyll Metabolism Pathway

3.1.1. Mutations of Genes Related to the Chlorophyll Synthesis Pathway

Chlorophyll in higher plants includes chlorophyll a and chlorophyll b. The biosynthesis of chlorophyll begins when L-Glutamyl-tRNA produces chlorophyll a, which is then oxidized by chlorophyllide, an oxygenase, to form the chlorophyll b synthesis cycle. This process involves a total of fifteen steps, and the whole synthesis process involves the participation of fifteen enzymes and the expression of twenty-seven genes encoding these enzymes (Figure 1 and Table 1) [37,38]. Any obstacle in this synthetic process will affect chlorophyll biosynthesis, resulting in leaf color mutants.

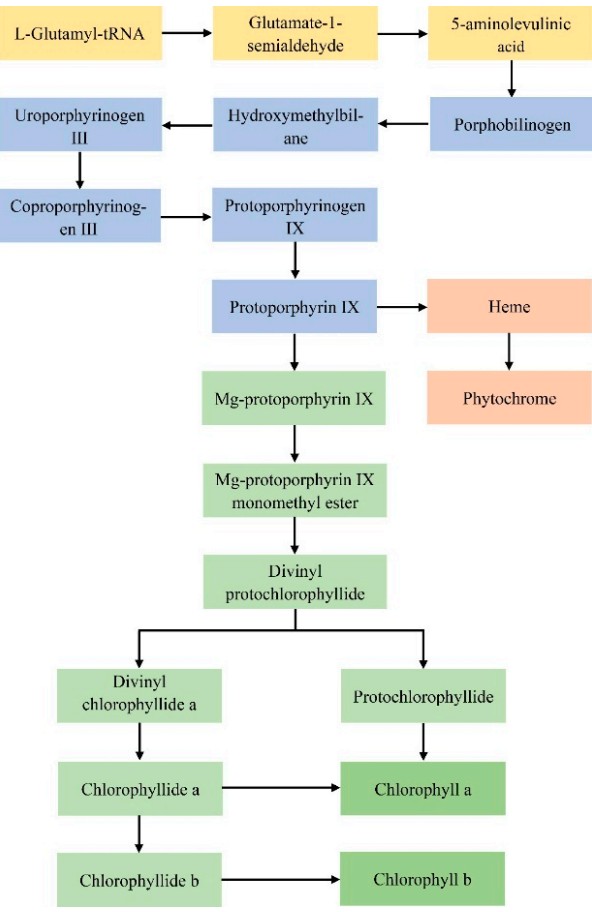

**Figure 1.** Biosynthetic pathway of angiosperm chlorophyll.

The entire process of chlorophyll synthesis is divided into two main parts: the first part includes the process of L-Glutamyl-tRNA synthesizing protoporphyrin IX (proto IX), and the second part encompasses from protoporphyrin IX to chlorophyll biosynthesis. The two synthetic parts can be divided into three stages: the first stage is from L-Glutamyl-tRNA to 5-aminolevulinic acid (ALA), the second stage is from ALA to proto IX biosynthesis, and the third stage is from proto IX to chlorophyll biosynthesis. Furthermore, the entire process of chlorophyll synthesis is localized at three tissues: the synthesis of ALA to proto IX is completed in the middle of the chloroplast stroma; the synthesis of proto IX to chlorophyllide occurs on the chloroplast membrane; and the process of chlorophyll a and chlorophyll b synthesis is completed on the thylakoid membrane [39]. Problems with any of genes in Table 1 may lead to changes in enzyme activity and function within chlorophyll synthesis process, resulting in the accumulation of excessive intermediates, affecting the normal metabolism of chlorophyll, and changing the proportion of pigments in chloroplasts, which can lead to oxidative damage, causing different plants to produce leaf color mutations or possibly even causing plant death [40]. For example, after glutamate-tRNA synthetase is silenced by the virus, the mutant phenotype is extremely yellow [41]. When the function of chlorophyll a oxidase is abnormal, chlorophyll b synthesis is reduced, or it is not synthesized [42]. In the fifteen steps of the chlorophyll metabolism pathway, the earlier the occurrence of the mutation, the more pronounced the leaf color mutations, which generally present as a yellow or white phenomenon. If the mutation occurs in the later stages of chlorophyll synthesis, it usually only results in patches, stripes, etc. [43].

In the chlorophyll biosynthesis pathway, the synthesis of ALA and the insertion of the Mg ion into proto IX are the two main control points that directly affect chlorophyll synthesis [44]. ALA is catalyzed by Glutamyl-tRNA reductase (GluTR) and Glutamate-1-semialdehyde 2,1-aminomutase (GSA-AM), which controls the rate of chlorophyll and heme synthesis. This is the rate-limiting step of

the tetrapyrrole synthesis pathway, and it plays a key role in chlorophyll synthesis [45]. For example, the overexpression of the *HEMA1* gene in yellowing plants leads to an increase in protochlorophyllide content [46]. The synthesis of ALA is influenced by GluTR reductase, and GluTR is encoded by the *HEMA* gene, of which plants contain at least two (*HEMA 1, HEMA 2*) [47,48]. The antisense *HEMA1* gene was transferred into *A. thaliana*, and it was found that the antisense *HEMA1* gene inhibited the formation of ALA, which resulted in decreases of protochlorophyllide synthesis and chlorophyll content [49]. The activity of GluTR is regulated by heme, which as a terminal product can also provide feedback regulation on the activity of GluTR. For example, gene mapping of the *A. thaliana* 'ulf' mutant was located at *Hy1* site, which encodes for the synthesis of heme oxidase, and the reduced heme oxidase activity in the mutant led to the accumulation of heme, which inhibited the activity of GluTR [50].

**Table 1.** Genes encoding and enzymes involved in the chlorophyll synthesis pathway in *A. thaliana*.

| Step | Enzyme Name | Abbreviated Name of Enzyme | Gene Name | Locus Name in *Arabidopsis* |
|---|---|---|---|---|
| 1 | Glutamyl-tRNA reductase | GluTR | *HEMA1* | AT1G58290 |
| | | | *HEMA2* | AT1G09940 |
| | | | *HEMA3* | AT2G31250 |
| 2 | Glutamate-1-semialdehyde 2, 1-aminomutase | GSA-AM | *GSA1 (HEML1)* | AT5G63570 |
| | | | *GSA2 (HEML2)* | AT3G48730 |
| 3 | 5-Aminolevulinate dehydratase | PBGS (ALAD) | *HEMB1* | AT1G69740 |
| | | | *HEMB2* | AT1G44318 |
| 4 | Porphobilinogen deaminase | PBGD | *HEMC* | AT5G08280 |
| 5 | Uroporphyrinogen III synthase | UROS | *HEMD* | AT2G26540 |
| 6 | Uroporphyrinogen III decarboxylase | UROD | *HEME1* | AT3G14930 |
| | | | *HEME2* | AT2G40490 |
| 7 | Coproporphyrinogen III oxidase | CPOX | *HEMF1* | AT1G03475 |
| | | | *HEMF2* | AT4G03205 |
| 8 | Protoporphyrinogen oxidase | PPOX | *HEMG1* | AT4G01690 |
| | | | *HEMG2* | AT5G14220 |
| 9 | Magnesium chelatase H subunit | | *CHLH* | AT5G13630 |
| | Magnesium chelatase I subunit | MgCh | *CHL11* | AT4G18480 |
| | | | *CHL12* | AT5G45930 |
| | Magnesium chelatase D subunit | | *CHLD* | AT1G08520 |
| 10 | Mg-protoporphyrin IX methyltransferase | MgPMT | *CHLM* | AT4G25080 |
| 11 | Mg-protoporphyrin IX monomethyl ester | MgPME | *CRD1 (ACSF)* | AT3G56940 |
| 12 | 3,8-Divinyl protochlorophyllide a 8-vinyl reductase | DVR | *DVR* | AT5G18660 |
| 13 | Protochlorophyllide oxidoreductase | POR | *PORA* | AT5G54190 |
| | | | *PORB* | AT4G27440 |
| | | | *PORC* | AT1G03630 |
| 14 | Chlorophyll synthase | CHLG | *CHLG* | AT3G51820 |
| 15 | Chlorophyllide a oxygenase | CAO | *CAO (CHL)* | AT1G44446 |

Magnesium chelatase is the key factor in guaranteeing the biosynthesis of chlorophyll, which can catalyze the formation of Mg protoporphyrin by combining a magnesian ion with protoporphyrin [51]. The reaction step of metal ion insertion into protoporphyrin IX is the branching point for the synthesis of chlorophyll, heme and plant pigments. Magnesium chelatase catalyzes magnesian ion insertion into protoporphyrin IX to form the chlorophyll branch, while ferric chelatase catalyzes ferrous ion to be inserted into protoporphyrin IX to form the heme and plant pigment branches. Magnesium chelatase and ferric chelatase compete for protoporphyrin IX at the branch point [45,52]. Studies have shown that the main reason for the formation of chlorophyll mutants is problems with magnesium chelatase, which prevents the synthesis of chlorophyll from proceeding normally [53]. The decrease in magnesium chelatase activity leads to a decrease in chlorophyll content in the mutant, and the phenotype of the mutant will also be affected by the enzyme activity [54]. The function of the magnesium chelatase is complex, which consists of the *D*, *I*, and *H* subunits, depends on the synergy of the three functional subunits. For example, in *Arabidopsis*, a single base mutation in the third exon of the *GUN5* gene encoding the magnesium chelatase *H* subunit, will cause the formation of albinos [55]. If the *CHL1* gene encoding the magnesium chelatase *D* subunit mutates, the mutant will present as a light yellow-green

at the seedling stage, and the expression of the nuclear gene *LHCP II* will also be affected through a feedback regulation mechanism [56]. In addition, chlorophyll synthesis may be regulated by other factors besides the expression of the twenty-seven gene encoding enzymes. For example, in a study of *Pak-choi* yellow leaf mutants, the expression of *CHLG* encoding chlorophyll synthase was found to be significantly decreased, but there was no difference between wild type and mutant *CHLG* sequences, indicating that the chlorophyll synthetase activity was also regulated by other regulatory factors [57].

### 3.1.2. Mutations of Genes Related to the Chlorophyll Degradation Pathway

In leaves with no color mutation, chlorophyll degradation is coupled with simultaneous chlorophyll synthesis in a dynamic balance. Chlorophyll degrades during the transformation of chloroplasts into chromoplast, which indicates that leaves will begin to age [58]. The reaction process of chlorophyll degradation is mainly divided into two stages. The first stage is the degradation of chlorophyll into the primary fluorescent chlorophyll catabolite (pFCC). In the second stage, the pFCC in the vacuole forms nonfluorescent chlorophyll catabolite (NCC), which is finally transformed into the oxidation degradation product, pyrrole (Figure 2) [59]. This degradation pathway involves a series of complex reactions, and an interruption in any step may change the chlorophyll content and produce leaf color mutations.

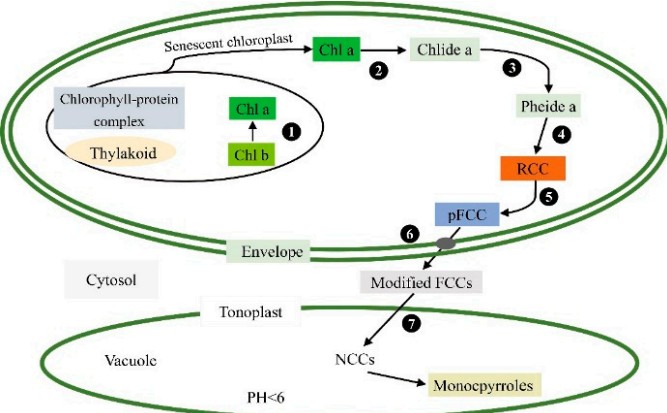

**Figure 2.** Biodegradation pathway of chlorophyll in higher plant. (1) Chl b reductase; (2) Chlorophyllase; (3) Metal chelating substance; (4) Pheide a oxygenase; (5) RCC reductase; (6) Catabolite transporter; (7) ABC transporter. Chl a, Chlorophyll a; Chl b, Chlorophyll b; Chlide a, chlorophyllide a; Pheide a, Pheophorbide a; RCC, red Chl catabolite; pFCC, primer fluorescent Chl catabolite; FCCs, fluorescent Chl catabolites; NCCs, nonfluorescent Chl catabolites.

Mutation of the chlorophyll degradation pathway will lead to a lack of chlorophyll degradation in plants, resulting in the stay-green phenomenon in plant leaves; this kind of mutation is called a stay-green mutation. In the *Z. mays 'fs854'* mutant, the inhibition of the chlorophyll degradation pathway led to the occurrence of the stay-green phenomenon in maize, which promoted the increase of maize yield due to its high photosynthesis [60]. If the process of chlorophyll degradation is accelerated, leaf color mutants may also be produced. The expression of *CHL2* and *RCCR* encoding key chlorophyll degradation enzymes in *Cymbidium sinense* mutants was higher than wild-type, resulting in a decrease in chlorophyll content, which may be the reason for the yellow color mutation of Moran leaf [61]. Stay-green mutations have a distinct phenotype that can delay the leaf senescence of crops and increase crop yield. It has a much higher potential application value compared with yellow-green seedlings, and it therefore attracts significant research attention.

### 3.1.3. Mutations of Genes Related to the Heme Metabolism Pathway

Tetrapyrrole is the skeleton structure and common precursor of chlorophyll and heme biosynthesis in plants, the biosynthetic pathway of which is divided into two branches, the chlorophyll biosynthetic pathway and the heme biosynthetic pathway, at protoporphyrin IX. The first branch chelates protoporphyrin IX with a magnesian ion to produce Mg-protoporphyrin IX, which finally forms chlorophyll; the second branch chelates proto IX with a ferrous ion to form heme [40]. Heme is a type of iron-containing cyclic tetrapyrrole that is the intermediate product of the synthetic pathway of phytochrome and phycobilin [62]. After a series of complex reactions, heme finally forms a photosensitive pigment chromophore (Figure 3) [40]. As heme and chlorophyll in plants have the same precursor and share part of the same pathway, they are the branch products of chlorophyll biosynthesis. Therefore, the synthesis of chlorophyll is regulated by the feedback inhibition of intracellular heme content. When the heme metabolism branch of plants is disturbed, it may lead to an increase in heme content in the cells. High concentrations of heme inhibit the synthesis of ALA through feedback regulation, which is a common precursor of chlorophyll and heme, resulting in the inhibition of chlorophyll synthesis and variations in leaf color [63].

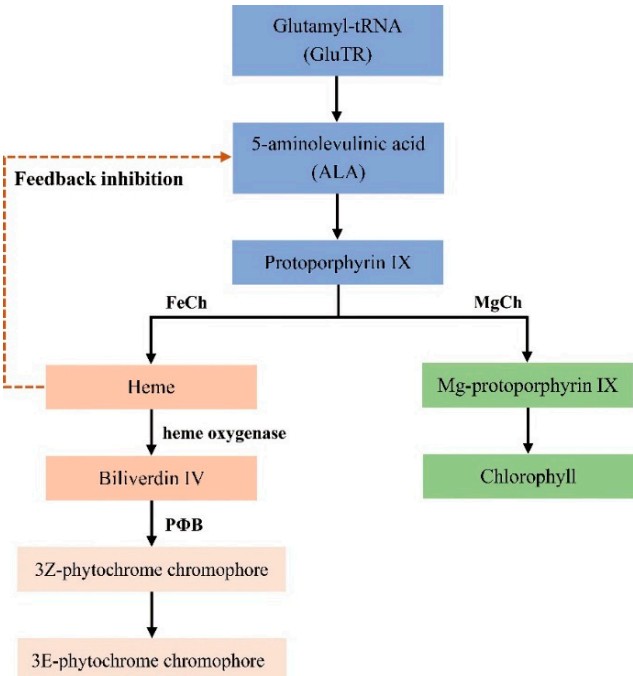

**Figure 3.** Heme metabolism pathway in high plants. The 3Z-phytochrome chromophore could spontaneously form a 3E-phytochrome chromophore or be an enzyme catalyst.

Heme oxygenase (HO) catalyzes the degradation of heme to synthesize phytochrome precursors in higher plants, which controls the rate of heme degradation. Both pea and *Arabidopsis* leaf color mutants are caused by *HO1* deficiency [64]. The *Arabidopsis 'flu'* mutant gene is located at the *Hy1* site, which encodes a heme oxidase. A decrease in heme oxygenase activity in the mutant results in the accumulation of heme, while the accumulation of heme inhibits the activity of glutamyl tRNA reductase, which indicates that heme can provide feedback regulation for the activity of glutamyl tRNA reductase [50]. In the study of the rice *YELLOW-GREEN LEAF2* mutant, a 7-kb insertion was found in the first exon of *YGL2/HO1*, resulting in a significant reduction in the expression of *ygl2* in the *ygl2* mutant, which hinders chlorophyll synthesis and leads to the yellow leaf phenotype [65]. A 45 bp fragment was found to be inserted into the gene encoding heme oxygenase through map-based cloning, which inhibited the expression of the heme oxygenase encoding gene and resulted in leaf

color mutation [66]. In addition, leaf color mutants lacking phytochrome chromophores can be used to study the regulation of heme on chlorophyll synthesis, as well as may be used as an ideal model with which to study the photomorphogenesis of higher plants [67].

## 3.2. Abnormal Chloroplast Development and Differentiation

Chloroplasts originate from protoplasts and are autonomous organelles in plant cells. Their differentiation and development are controlled by the interaction and co-regulation of nuclear genes that encode related proteins and plastid genes involving gene transcription, RNA processing, and protein translation, folding and transportation [68]. This process can be divided into seven steps: nuclear gene transcription, chloroplast protein input and processing, chloroplast gene transcription and translation, thylakoid formation, pigment synthesis, plasmid–nuclear signal transduction, and chloroplast division [69]. Additionally, chloroplast development is closely related to chlorophyll content, and any obstructed pathway in the process of chloroplast development leads to chloroplast hypoplasia, thus affecting chlorophyll content in plants and causing leaf color mutations.

The regulatory pathway of the sesame yellow leaf character mutation was first analyzed in the '*Siyl-1*' sesame mutant with yellow-green leaf color. The results showed that the number of chloroplasts and the morphological structure of the mutants changed significantly, and the chlorophyll content also decreased significantly [33]. The generation of leaf color variation is also closely related to the obstruction of the plastid-nuclear signal transduction pathway. Nuclear genes can encode chloroplast proteins, regulate the metabolic state of chloroplasts, and regulate the transcription and translation of chloroplast genes. Plastids can also regulate nuclear gene expression via retrograde signaling pathways [70]. In the study of *Arabidopsis* '*cue*' mutants, the leaves of '*cue*' mutants with the visible phenotypes (virescent, yellow-green, pale), defective chloroplast development, delayed differentiation of chloroplasts, and defects in mesophyll structure. The results show that there were complex gene interactions in the cells, and nuclear-cytoplasmic genes coordinated the growth and development of plants through signal molecules [71]. The dynamic balance of chloroplast development, protein synthesis, and degradation is also an important factor affecting leaf color mutation [72]. The results of rice pale green mutants indicated that the decrease of chlorophyll content in leaves was mainly caused by the mutation of the protein *CSP41b* encoding chloroplast development [73].

In addition to being an important part of chloroplast structure, the thylakoid also plays an important role in the function of chloroplasts. One study showed that variation in thylakoid and vesicle structure can lead to the formation of '*vipp1*' in *Arabidopsis* mutants [74]. While researching the barley yellow-green leaf color mutant '*ygl9*', it was found that grana and stroma thylakoids were severely linear, grana could not be stacked normally, and the chloroplast ultrastructure was damaged in leaves of '*ygl9*' at seeding stage. [75]. In the study of a watermelon *yellow leaf (YL)* mutant, compared with ZK, the chloroplasts of the YL plant underwent incomplete development, resulting in a small number of grana thylakoids in the chloroplasts in which the arrangement was disordered and the cell metabolism was weak; these changes ultimately affected the light harvesting ability of the plants [76].

## 3.3. Abnormal Carotenoid Metabolism Pathway

Carotenoids are mainly divided into yellow carotene and orange lutein. Carotenoids have the function of absorbing and transmitting light energy in plants, which can play an important role in the protection of chlorophyll [77]. Regulation of carotenoid biosynthesis is mainly achieved through regulation of carotenoid content via the level of transcription of the enzymes and genes involved in carotenoid synthesis, and through regulating the type and quantity of carotenoids produced [78].

The main genes involved in plant carotenoid synthesis are phytoene synthase (PSY), phytoene desaturase (PDS), ζ-carotene desaturase (ZDS), lycopeneβ-cyclase (LCYB) and lycopene epsilon-cyclase (LYCE), among others. Phytoene synthase is considered the most important regulatory enzyme in the carotenoid synthesis pathway [79]. In *Arabidopsis*, the overexpression of *PSY* can lead to a significant increase in β-carotene content in leaves, and a corresponding increase in total carotenoids,

which facilitates greening when etiolated seedlings emerge from the soil [80]. Ectopic expression of *PSY1* in tobacco leads to leaves showing abnormal pigmentation, and very young leaves are sometimes colored bright orange, before rapidly turning green [81]. The *immutans (im)* variegation mutant of *Arabidopsis* has light-dependent variegated phenotypes (green and white leaf sectors), the green leaf sectors contain normal chloroplasts, while the white leaf sectors contain abnormal chloroplasts that lack carotenoids due to a defect in phytoene desaturase activity [82,83]. In the study of *PHS* mutants in rice, the main enzymes involved in the anabolic pathway of carotene and lutein were mutated, resulting in photooxidative damage of leaf photosystem II (PSII). The core proteins CP43, CP47, and D1 of PSII were all reduced, accompanied by the accumulation of reactive oxygen species (ROS) in the plants. This result showed that damage to carotenoid biosynthesis would result in photooxidation damage and abscisic acid (ABA) deficiency, which would lead to the sprouting of spikes and the appearance of albinism in leaves [84]. Park et al. [85] showed that carotenoids have a feedback regulating effect on plastids, such that changes in carotenoid synthesis can regulate the development of plastids.

### 3.4. Abnormal Anthocyanin Metabolism Pathway

Anthocyanins are a class of water-soluble pigments widely present in plants that belongs to a group of flavonoids produced by the secondary metabolism of plants. It mainly accumulates in the form of glycosides in plant vacuoles and can be transformed from chlorophyll. The generalized term anthocyanin refers to all kinds of anthocyanin glycosides [86]. The anthocyanin biosynthesis pathway is an important branch of the flavonoid metabolism pathway, which is closely related to the presentation of plant color [87]. Its biosynthetic pathway has been clearly demonstrated through the research of model plants such as *Arabidopsis*, corn, petunia, etc.

### 3.4.1. Mutations of Structural Genes Related to the Anthocyanin Synthesis Pathway

Anthocyanin biosynthesis is a direct precursor of phenylalanine in the cytoplasm, which is catalyzed by a series of enzymes encoded by structural genes. After various modifications, it is transported to the vacuole and other places for storage. The synthetic pathway from phenylalanine to anthocyanins can be divided into three stages: (1) phenylalanine produces 4-coumaroyl CoA, which is catalyzed by phenylalanine ammonia-lyase; (2) 4-coumaroyl CoA produces colorless anthocyanins, which are catalyzed by biological enzymes such as chalcone synthase; and (3) colorless anthocyanins catalyze the production of anthocyanins through a series of enzymes in the end-stage metabolic pathway. In this process, a series of structural, modification and transport-related genes are directly and positively regulated by a complex (MBW complex for short) composed of MYB (v-myb avian myeloblastosis viral oncogene homolog), bHLH (basic helIX-loop-helIX) and WDR (WD repeat) regulatory factors [88,89]. Therefore, the genes that affect anthocyanin metabolism can be divided into two categories: structural genes and regulatory genes. Anthocyanin synthesis in plants is strongly influenced by the structural genes in the metabolic pathway, because the structural genes directly encode the enzymes needed in the biosynthesis pathway of anthocyanin metabolism, including phenylalanine ammonia-lyase (PAL), chalcone synthase (CHS), chalcone isomerase (CHI), flavanone 3'–5' hydroxylase (F3'5'H) and anthocyanin synthase (ANS), etc. (Figure 4) [90,91]. Variations in these structural or regulatory genes can lead to the formation of various leaf color mutations.

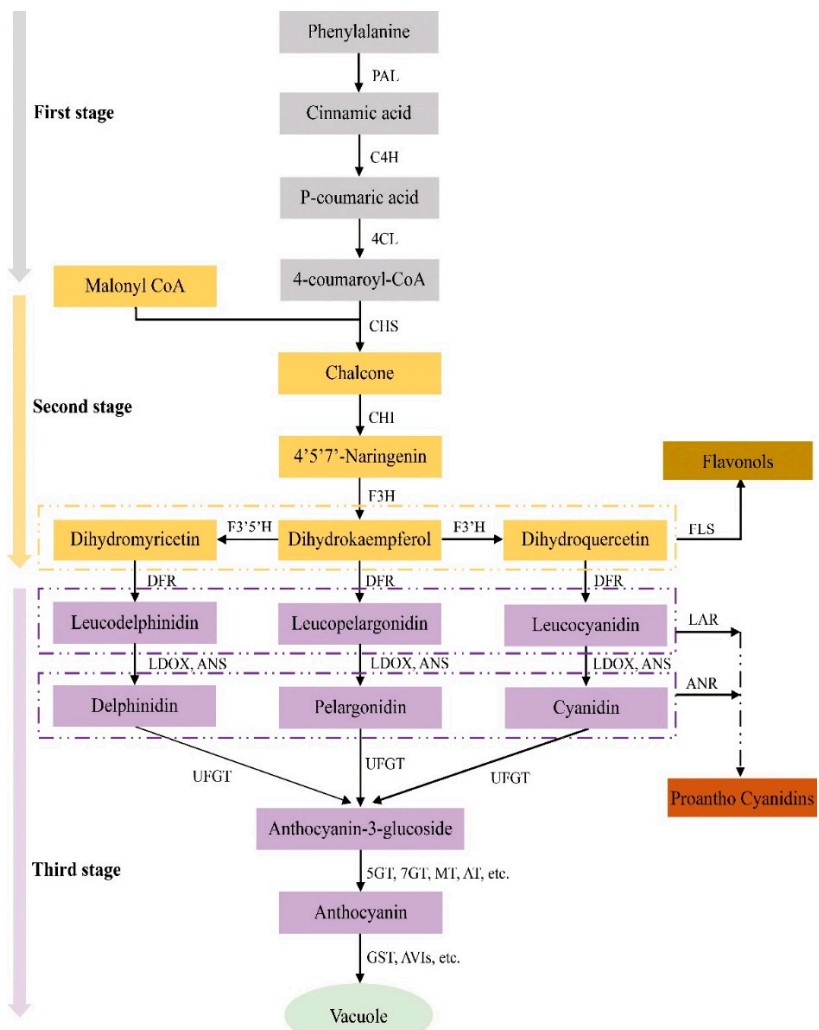

**Figure 4.** Biosynthesis pathway of anthocyanins in plants. PAL, Phenylalanine ammonia lyase; C4H, Cinnamate 4-hydroxylase; 4CL, 4-coumarate CoA ligase; CHS, Chalcone synthase; CHI, Chalcone isomerase; F3H, Flavanone 3-hydroxylase; F3′H, Flavonoid 3′-hydroxylase; F3′5′H, Flavonoid 3′,5′-hydroxylase; DFR, Dihydroflavonol 4-reductase; ANS, Anthocyanidin synthase; LDOX, leucoanthocyanidin dioxygenase; UFGT, Flavonoid 3-O-glucosyltransferase; 5GT, Anthocyanin 5-O-glucosyltransferase; 7GT, Flavonoid 7-O-glucosyltransferase; MT, Methyl transferase; AT, Anthocyanin acyltransferase; GST, Glutathione S-transferase; AVIs, Anthocyanic vacuolar inclusions.

In the anthocyanin metabolism pathway, the PAL is the initial enzyme of anthocyanin synthesis and an important regulatory site. Its expression is regulated by its own development and environmental factors. The PAL activity and anthocyanin content of purple-foliage plum leaves were found to increase under light induction, and the leaves gradually turned purplish red [92]. CHS is another key enzyme in the anthocyanin biosynthetic pathway. By using RNA interference to inhibit the expression of *Torenia hybrida CHS*, the blue *T. hybrida* rich in mallow pigment and peony pigment can be transformed into white *T. hybrida* with anthocyanin deficiency [93]. When *Freesia hybrida CHS1* was imported into *Petunia hybrida*, its color changed from white to pink [94]. CHI is also related to the performance of plant leaf color. Inhibiting its expression or reducing its enzyme activity causes a large accumulation of 4′-hydroxychalcone in the anthocyanin synthesis pathway, which affects the rate of anthocyanin synthesis [95]. For example, *CHI* silencing can turn the color of carnation and tobacco to yellow [96]. Moreover, the ANS regulates the oxidation of downstream colorless proanthocyanin into colored anthocyanin. In a study of the onion mutant *ANSPS*, the gene encoding the ANS was found to

have a base insertion mutation, which caused the mutant material to have a different yellow color from the wild type [97]. Overexpression of the *F3′5′H* in roses and chrysanthemums can lead to the synthesis of delphinium in flowers, thus generating blue-hued flowers [98,99]. Therefore, disorders of the anthocyanin metabolic pathway are an important mechanism through which to produce leaf color mutations.

### 3.4.2. Abnormal Regulatory Factors Related to the Anthocyanin Synthesis Pathway

Anthocyanin synthesis regulatory genes play an important role in plant metabolism. Although they are not structural genes, they are very important for the regulation of structural genes. These transcription factors regulate the metabolic pathways of anthocyanin biosynthesis by binding to the cis-acting elements in the promoters of structural genes to regulate the expression of one or more genes in the biosynthesis pathway. There are three types of transcription factors: MYB (v-myb avian myeloblastosis viral oncogene homolog), bHLH (basic helIX-loop-helIX), and WDR (WD40 repeat proteins). The expression of structural genes is directly controlled by the MBW (MYB-bHLH-WD40) protein complex, which is formed by three types of transcription factors: MYB, bHLH and WDR. The specific combinations of MYB, bHLH and WDR in the MBW protein complex determine the target and intensity of the complex regulation [100,101].

In the process of anthocyanin biosynthesis, most *MYB* transcription factors have positive regulatory effects and usually exist in plant genomes in the form of gene families, including two motifs, such as *R2* and *R3*, or one motif in *R3*. The helIX-helIX-turn-helIX structure at the C-terminal of the conserved domain can specifically recognize and bind to DNA, while its highly conserved C-terminal is of great significance for anthocyanin regulation [102,103]. *PtrMYB119* function as transcriptional activators of anthocyanin accumulation in both Arabidopsis and poplar. Overexpression of *PtrMYB119* in hybrid poplar resulted in elevated accumulation of anthocyanins in whole plants, which had strong red-color pigmentation, especially in leaves relative to non-transformed control plants [12]. Additionally, *PAP1* (production of anthocyanin segment 1) is a key *MYB* class transcription factor that regulates anthocyanin synthesis and affects the expression of structural genes such as *CHS*, *CHI* and *ANS*. For example, activating the overexpression of *Arabidopsis PAP1* can induce the accumulation of anthocyanins in large quantities, which leads to the dark purple color of the leaves of *Arabidopsis PAP1* [104]. In blood-fleshed peach, the coloring BL upstream of the MYB can activate the anthocyanin MYB transcription, resulting in anthocyanin accumulation in the flesh, which then becomes blood-colored [105].

The bHLH transcription factor is involved in the regulation of various physiological pathways, among which the regulation of flavonoid and anthocyanin synthesis is one of its most important functions [106]. In *Setaria italica*, *PPLS1* is a bHLH transcription factor that regulates the purple color of pulvinus and leaf sheath in foxtail millet. Co-expression of both *PPLS1* and *SiMYB85* in tobacco leaves increased anthocyanin accumulation and the expression of genes involved in anthocyanin biosynthesis (*NtF3H, NtDFR* and *Nt3GT*), resulting in purple leaves [107]. Moreover, the bHLH transcription factor can also coregulate the target gene with MYB. For example, expression of *Medicago truncatula MtTT8* in *A. thaliana tt8* mutant can produce anthocyanins and restore the phenotype of anthocyanin deficiency of the mutant [108]. The regulatory R2R3-MYB transcription factor *AmRosea1* and the bHLH transcription factor *AmDelila* in *Antirrhinum majus* have been extensively studied. The simultaneous expression of the *AmRosea1* and *AmDelila* activates anthocyanin production in the orange carrot, which young leaves were dark purple and matured into dark green leaves during the subsequent growth relative to non-transformed plants [109]. In addition, the chrysanthemum *CmbHLH* transcription factor can activate *DFR* expression and promote anthocyanin synthesis when co-expressed with the *CmMYB6* transcription factor [110].

In all the WDR proteins studied, their functions were more similar to those of an interaction platform between other proteins, and their role may that of an intermediate medium connecting MYB and bHLH to form a complex [111]. For example, the *bHLH* transcription factors *TT8*, *EGl3* and WD40 repeat proteins *TTG1* of *Arabidopsis* form MBW complexes, which regulate the expression of

*DFR*, *ANS* and *tt19*, thus affecting anthocyanin biosynthesis [112]. In addition, WD40 can stabilize the MBW protein complex and directly regulate the transcriptional regulation of the anthocyanin synthesis pathway [113]. For example, in the study of *M. truncatula*, the silencing of *MYB5* and *MYB1* in *Arabidopsis* was found to directly change the expression pattern of the structural genes, resulting in a decrease in anthocyanin content [114].

## 4. Mapping and Cloning of Leaf Color Mutational Genes

With the rapid development of next-generation genome sequencing technologies, the application of genome, pangenome and transcriptome is increasingly extensive in revealing the genetic variation traits and dissecting the genetic regulation mechanism of key genes in plants. Current research is focused on revealing the mechanisms of growth, fruit quality, stress response, and mutation traits, as well as gender determination and flowering, in plants [115–119]. Currently, map-based cloning and RNA-seq are mainly used to mine genes related to leaf color mutations. By using next-generation genome sequencing technologies for the analysis of genomic and transcriptomic data, it is possible to greatly improve the efficiency and precision of the identification of leaf color mutation related genes in plant. The identification, cloning and functional study of these mutant genes plays an important role in elucidating the molecular mechanisms of leaf color mutation, plant photosynthesis, pigment synthesis, etc.

### 4.1. Using Map-Based Cloning to Discover Mutant Genes

Map-based cloning, also known as positional cloning, is a new technology for identifying and isolating genes through forward genetics. This technology can gradually locate and clone the gene of interest based on the traits of the organism and the position of closely linked molecular markers on the chromosome without knowing the sequence of the gene of interest in advance. Map-based cloning technology includes the following three steps: the first is to build positioning groups through inbreeding, hybridization, or back-crossing, etc.; the second is to find the markers linked with the gene of interest through molecular marker technology to build a genetic map; and the third is to predict and screen candidate genes and determine the function of the gene of interest by means of molecular biology [120].

The genes controlling leaf colors in plants are usually closely related to genes related to chlorophyll metabolism or chloroplast development. For instance, in the study of rice mutants, gene mapping of yellow-green leaf color genes in rice were performed using map cloning. The results showed that the gene encoding the chloroplast signal recognition particle had a mutation from *A* to *T*, which led to the change in leaf color [121]. In the study of tomato mutants, a single base mutation in the first intron was found using map-based cloning to result in downregulation of *WV* expression, which inhibited the expression of chloroplast-encoded genes and blocked chloroplast formation and chlorophyll synthesis, thereby producing leaf color mutants [122]. At present, map-based cloning technology is increasingly being used to study mutants, and many genes related to chlorophyll synthesis and chloroplast development have been cloned in rice, barley, pear, *Brassica oleracea*, and cucumber [23,123–126].

### 4.2. Using RNA-Seq to Discover Mutant Genes

RNA-seq, also known as mRNA-seq, is used to study changes in gene expression at the mRNA level and can directly perform transcriptome analysis on a species without reference to genome information. At present, it is considered to be an effective method for discovering new genes and for annotating coding and noncoding genes in the study of plant leaf color mutants, based on the different expression levels of transcripts among different samples to obtain the differentially expressed genes (DEGs) [127]. Through the functional annotation and enrichment analysis of differentially expressed genes, we can discover some key candidate genes, which is helpful in clarifying the molecular mechanisms of leaf color mutation [128].

RNA-seq is widely used in leaf color mutants of ornamental plants. For example, in the mutants (etiolated, rubescent, and albino) of *Anthurium andraeanum* 'Sonate', the ultrastructure of chloroplasts in mutant leaves was disrupted, and very few intact chloroplasts were observed. The ratio of carotenoid/total Chl in all three mutants was higher than that of the wild type and the content of anthocyanin was at a similar level in the leaf of etiolated mutant and wild type plants, while the albino leaves had the lowest anthocyanin content. After RNA sequencing, most genes related to chlorophyll synthesis, anthocyanin transport and pigment biosynthesis were down-regulated compared with the wild type. These results indicate that the abnormal leaf color of mutants was caused by changes in the expression pattern of genes responsible for pigment biosynthesis [129]. Therefore, it can be concluded that the chloroplast development, division and the mutation of chlorophyll and anthocyanin synthesis pathway in anthocyanin mutants affected the synthesis of chlorophyll and anthocyanin, and finally led to abnormal leaf color [130]. In addition, we also found the main differentially expressed genes related to chlorophyll metabolism, chloroplast development and the photosynthetic system after analyzing the transcriptome data of cucumber [32], tomato [131], barley [132], wheat [133], Birch [134], etc.

## 5. Conclusions

A growing number of genes that control or influence pigment metabolism, chloroplast development and differentiation, and photosynthesis have been identified from a variety of leaf color mutants. Through the study of their function and interaction relationships, the formation mechanism of leaf color mutants has been clarified, which is of great significance to the study of leaf color mutants. At present, research on the formation of leaf color mutants has focused on the functions of related genes and transcription factors, such as chlorophyll metabolism, chloroplast development and anthocyanin metabolism, while few studies have examined nuclear-cytoplasmic gene interactions, transcription factors, regulatory elements, and other pigment metabolism. Moreover, gene transcription and translation are influenced not only by transcription factors but also by non-coding RNA (such as miRNA and lncRNA), epigenetics (such as DNA methylation and histone acetylation), and protein spatial structure, although few studies have reviewed these influences. Therefore, in future studies, we should strengthen the research on other pigment molecules, nuclear-plasmid signal transduction, and transcription regulation, as well as examining noncoding RNA, DNA methylation, etc. This will help us to obtain more relevant information on the regulation of pigment anabolic metabolism, and to analyze the molecular regulation mechanisms of plant leaf color mutations from a system network level.

**Author Contributions:** Conceptualization, X.-Y.Z. and M.-H.Z.; methodology, X.L., X.-X.Z. and H.Z.; validation, M.-H.Z., X.L. and X.-X.Z.; resources, M.-H.Z. and H.Z.; writing—original draft preparation, M.-H.Z.; writing—review and editing, R.R.-S. and X.-Y.Z.; supervision, X.-Y.Z.; project administration, X.-Y.Z.; funding acquisition, X.-Y.Z. All authors have read and agreed to the published version of the manuscript.

**Funding:** This research was funded by the Fundamental Research Funds for the Central Universities, grant number 2572017DA02.

**Conflicts of Interest:** The authors declare no conflict of interest.

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
