# Peer review of "Mutation Mechanism of Leaf Color in Plants: A Review"

_forests, doi:10.3390/f11080851_

Round 1

Reviewer 1 Report

The Zhao's paper is the review of mutation mechanisms of leaf color in plant. They explained first genetic model of plant leaf color mutation, and then described the molecular mechanisms of plant leaf color mutations related to chlorophyll metabolic pathways, chloroplast development and differentiation, carotenoid and anthocyanin metabolic pathways. Finally, they summarized the mapping and cloning of leaf color mutational genes.

This is a well-written review covered many aspects of leaf color mutation. For the reader's convenience, some clarifications and alterations are recommended. In particular, it seems that the carotenoid and anthocyanin parts (line 256 - 384) include descriptions of genes or metabolisms that are not related to leaf color mutants directly. Please focus on the leaf color mutants, and edit the related parts.

Followings are specific comments on the manuscript.

1) Abstract. line 12 and 13. The meanings of "directly" and "social society" are not clear. Please clarify them.

2) Line 28. What is "material carrier"? Please rewrite this sentence.

3) Line 82. studied => summarized

4) Line 89 - 90. Please check the values. It seems that the steps are 15, the genes are 26, and the enzymes are 17.

5) Table 1. Step 4. porphobilinogen => Porphobilinogen

6) Line 93 - 102. This part is complicated. Please describe the standard of the classification of the process, and simplify the classification.

7) Line 102. What are the meanings of "these genes"? Maybe "these genes" => "genes in the Table 1".

8) Line 103. Please clarify "this reaction process". Which reaction process do the authors suggest?

9) Line 108. What is the meaning of "chlorophyll an oxidase function is abnormal"?

10) Line 109 - 110. What is the meaning of "the earlier the mutation sites occurred"?

11) Figure 1. Please explain the meanings of the different colors.

12) Line 145. Please add "is" before "complex".

13) Line 158. normal plants => the leaves with no color mutation,
accompanied by => coupled with.

14) Line 159 - 160. What is the meaning of this sentence?

15) Line 176. obvious => distinct

16) Line 176. "delay the leaf senescence of crops" may be better than "prolong the maturity period of crops".

17) Line 177. It has a much higher potential application value compared with ...

18) Line 198. ...plants is disturbed, it may lead to an increase in ...

19) Line 214 - 216. ..phytochrome chromophores can be used to study the ...synthesis, as well as may be used as an ideal model ...

20) Line 222. Please clarify the meaning of "common".

21) Line 225. seven steps => six steps?

22) Line 238 - 239. Nuclear genes closely related to what?

23) Line 239 - 241. How about the chloroplast development in the 'cue' mutant?

24) Line 249 - 250. Please clarify the meaning of this sentence. What are "basal" and "linearized"?

25) Line 257 - 258. It is better to delete this sentence.

26) Line 257 - 280. It is not clear how carotenoid metabolism pathway is related to color mutation.

27) Line 355 - 356. This mutant may be of fruit, not of leaves.

28) Line 365 - 369. These mutants may be of flowers, not of leaves.

29) Line 427. Please add the results for carotenoid and anthocyanin reported in [122].

Author Response

1) Abstract. line 12 and 13. The meanings of "directly" and "social society" are not clear. Please clarify them.

Response 1): Abstract. line12 and 13. directly => unwittingly. Delete “a social”

2) Line 28. What is "material carrier"? Please rewrite this sentence.

Response 2): Line 28. “As the material carrier of photosynthesis…” => “As the part of photosynthesis …”

3) Line 82. studied => summarized

Response 3): Line 82. studied => summarized

4) Line 89 - 90. Please check the values. It seems that the steps are 15, the genes are 26, and the enzymes are 17.

Response 4): Line 89 - 90, nineteen => fifteen. 

After checked, the steps are 15, the enzymes are 15, the genes are 27. In the step 9, the “Magnesium chelatase H subunit, Magnesium chelatase I subunit, Magnesium chelatase D subunit” are belong to “Magnesium chelatase”, So they should be an enzyme.

5) Table 1. Step 4. porphobilinogen => Porphobilinogen

Response 5): Table 1. Step 4. porphobilinogen => Porphobilinogen

6) Line 93 - 102. This part is complicated. Please describe the standard of the classification of the process, and simplify the classification.

Response 6): Line 93 - 103. “Finally, the entire process of synthesis is divided into three parts…” => “Furthermore, the entire process of chlorophyll synthesis is localized at three tissues…”

The standard of the classification of the process is as follow:

The chlorophyll synthesis is divided into two main parts, the two parts (chlorophyll synthesis) are divided into three stages. The entire process of chlorophyll synthesis (two parts or three stages) is located at three tissues.

The synthesis pathway from L-Glutamyl-tRNA to Protoporphyrin IX (proto IX) is a common pathway for synthesis of other tetrapyrrole substances (such as heme) in plants [1], so the synthesis of chlorophyll is divided into two main parts.

The 5-aminolevulinic acid (ALA) is catalyzed by Glutamyl-tRNA reductase (GluTR), heme as a terminal product can provide feedback regulation on the activity of GluTR. ALA is the rate-limiting step of the tetrapyrrole synthesis pathway, and controls the rate of chlorophyll and heme synthesis [1]. Furthermore, the reaction step of metal ion insertion into protoporphyrin IX is the branching point for the synthesis of chlorophyll, heme and plant pigments. Therefore, the synthesis of ALA and protoporphyrin IX are the two main control points that directly affect chlorophyll synthesis [2,3], and the two synthetic parts (chlorophyll synthesis process) can be divided into three stages.

The synthesis process of chlorophyll is performed in chloroplast stroma, on chloroplast membrane and thylakoid membrane, so it is located at three tissues.

References:

  1. Cornah, J.E.; Terry, M.J.; Smith, A.G. Green or red: what stops the traffic in the tetrapyrrole pathway? Trends in Plant Science 2003, 8, 224-230, doi:10.1016/s1360-1385(03)00064-5.
  2. Walker, C.J.; Willows, R.D. Mechanism and regulation of Mg-chelatase. The Biochemical journal 1997, 327 ( Pt 2), 321-333.
  3. Strand, A.; Asami, T.; Alonso, J.; Ecker, J.R.; Chory, J. Chloroplast to   nucleus communication triggered by accumulation of Mg-protoporphyrinIX. Nature 2003, 421, 79-83, doi:10.1038/nature01204.

7) Line 102. What are the meanings of "these genes"? Maybe "these genes" => "genes in the Table 1".

Response 7): Line 103. these genes => genes in the Table 1.

8) Line 103. Please clarify "this reaction process". Which reaction process do the authors suggest?

Response 8): Line 104. this reaction => chlorophyll synthesis

9) Line 108. What is the meaning of "chlorophyll an oxidase function is abnormal"?

Response 9): Line 109 - 110. “When chlorophyll an oxidase function is abnormal…” => “When function of chlorophyll a oxidase is abnormal…”

The meaning of "chlorophyll a oxidase function is abnormal" is that When function of chlorophyll a oxidase is abnormal. For example, when the gene encoding chlorophyll a oxidase is mutated, the function of chlorophyll a oxidase will be abnormal.

10) Line 109 - 110. What is the meaning of "the earlier the mutation sites occurred"?

Response 10): In the fifteen steps of chlorophyll synthesis pathway, the mutation occurred farther forward, the more significant the effect of the mutation.

Line 111 - 112. Add “fifteen steps of” after “In the…”. Delete “sites”

11) Figure 1. Please explain the meanings of the different colors.

Response 11): The different colors represent three different stages of chlorophyll synthesis:

The yellow represents the first stage of chlorophyll synthesis that L-Glutamyl-tRNA to 5-aminolevulinic acid (ALA).

The blue represents the second stage of chlorophyll synthesis that ALA to proto IX biosynthesis.

The light green represents the third stage of chlorophyll synthesis that proto IX to chlorophyll biosynthesis.

The dark green represents the final products of chlorophyll synthesis, chlorophyll a and chlorophyll b.

The orange represents the heme synthesis.

12) Line 145. Please add "is" before "complex".

Response 12): Line 148. Add "is" before "complex".

13) Line 158. normal plants => the leaves with no color mutation,

accompanied by => coupled with.

Response 13): Line 161 - 162. normal plants => the leaves with no color mutation.

accompanied by => coupled with.

14) Line 159 - 160. What is the meaning of this sentence?

Response 14): Line 161 – 163 chromatids => chromoplast

Chlorophyll degrades during the transformation of chloroplasts into chromoplast, which indicates that leaves will begin to age.

15) Line 176. obvious => distinct

Response 15): Line 180. an obvious => a distinct

16) Line 176. "delay the leaf senescence of crops" may be better than "prolong the maturity period of crops".

Response 16): Line 180 - 181. "prolong the maturity period of crops" => "delay the leaf senescence of crops"

17) Line 177. It has a much higher potential application value compared with ...

Response17): Line 181 - 182. “It has higher a potential application value compared with …” => “It has a much higher potential application value compared with ...”

18) Line 198. ...plants is disturbed, it may lead to an increase in ...

Response 18): Line 203 - 204. “…plants is disturbed and abnormal, it leads to an increase in …” => “… plants is disturbed, it may lead to an increase in …”

19) Line 214 - 216. ..phytochrome chromophores can be used to study the ...synthesis, as well as may be used as an ideal model ...

Response 19): Line 220 - 223. “…phytochrome chromophores cannot only be used to study the …synthesis, but may also be an ideal model…” => “...phytochrome chromophores can be used to study the ...synthesis, as well as may be used as an ideal model ...”

20) Line 222. Please clarify the meaning of "common".

Response 20): Line 230. common regulation => co-regulation

21) Line 225. seven steps => six steps?

Response 21): Line 232 - 234. “…thylakoid formation and pigment synthesis…” => “…thylakoid formation, pigment synthesis…”

Therefore, this process can be divided into seven steps.

22) Line 238 - 239. Nuclear genes closely related to what?

Response 22): Line 242 - 248. plasma => plastid.

Line 245 - 248. “The development and metabolism of chloroplasts can also use feedback regulation to control closely related nuclear genes” => “Plastids can also regulate nuclear gene expression via retrograde signaling pathways”

In this study, it mainly refers to nuclear genes closely related to the development and metabolism of chloroplasts. Plastids can regulate nuclear gene expression via retrograde signaling pathways [1-2], such as GUN1-ABI4 pathway integrates multiple chloroplast-derived signals, which plays a role in the Mg-protoPIX pathway [3].   

Three independent plastid-to-nucleus retrograde signaling pathways have been studied [4-5]. In the best-defined pathway, accumulation of Mg–protoporphyrin IX (Mg-protoPIX), a chlorophyll biosynthetic intermediate, leads to down-regulation of hundreds of genes in Arabidopsis and has been shown to be involved in gene regulation in Chlamydomonas reinhardtii. Gene products of four Arabidopsis GENOMES UNCOUPLED (GUN) loci, GUN 2, 3, 4, and 5, are involved in modulating Mg-protoPIX levels. A second pathway represses Lhcb expression in response to inhibition of plastid gene expression (PGE) and requires GUN1 [4-5]. A second pathway represses Lhcb expression in response to inhibition of plastid gene expression (PGE) and requires GUN1. The third signaling pathway mediates signals de- rived from the reduction/oxidation (redox) state of the photosynthetic electron transfer chain (PET) and affects both photosynthesis-related and stress-related genes.

References:

  1. Kakizaki, T.; Matsumura, H.; Nakayama, K.; Che, F.-S.; Terauchi, R.; Inaba, T. Coordination of Plastid Protein Import and Nuclear Gene Expression by Plastid-to-Nucleus Retrograde Signaling. Plant Physiology 2009, 151, 1339-1353, doi:10.1104/pp.109.145987.
  2. Barajas-Lopez, J.d.D.; Blanco, N.E.; Strand, A. Plastid-to-nucleus communication, signals controlling the running of the plant cell. Biochimica Et Biophysica Acta-Molecular Cell Research 2013, 1833, 425-437, doi:10.1016/j.bbamcr.2012.06.020.
  3. Koussevitzky, S.; Nott, A.; Mockler, T.C.; Hong, F.; Sachetto-Martins, G.; Surpin, M.; Lim, I.J.; Mittler, R.; Chory, J. Signals from chloroplasts converge to regulate nuclear gene expression. Science 2007, 316, 715-719, doi:10.1126/science. 1140516.
  4. Gray, J.C. Chloroplast-to-nucleus signalling: a role for Mg-protoporphyrin. Trends Genet. 2003, 19, 526-529, doi:10.1016/j.tig.2003.08.001.
  5. Nott, A.; Jung, H.S.; Koussevitzky, S.; Chory, J. Plastid-to-nucleus retrograde signaling. In Annual Review of Plant Biology, Annual Reviews: Palo Alto, 2006; Vol. 57, pp. 739-759.

23) Line 239 - 241. How about the chloroplast development in the 'cue' mutant?

Response 23): The leaves of 'cue' mutants with the visible phenotypes (virescent, yellow-green, pale), defective chloroplast development, delayed differentiation of chloroplasts, and defects in mesophyll structure.

Line 248 - 254. “…it was found that there were complex gene interactions in the cells, and nuclear-cytoplasmic genes coordinated the growth and development of plants through signal molecules” => “… the leaves of 'cue' mutants with the visible phenotypes (virescent, yellow-green, pale), defective chloroplast development, delayed differentiation of chloroplasts, and defects in mesophyll structure. The result show that there were complex gene interactions in the cells, and nuclear-cytoplasmic genes coordinated the growth and development of plants through signal molecules”

24) Line 249 - 250. Please clarify the meaning of this sentence. What are "basal" and "linearized"?

Response 24): Line 261 - 267. basal => grana and stroma. linearized =>linear

“While researching the barley yellow green leaf color mutant ‘ygl9’, the basal thylakoids were severely were found to be linearized and the chloroplast ultrastructure was damaged in the seedling leaves of mutant ‘ygl9’” => “While researching the barley yellow-green leaf color mutant ‘ygl9’, found that grana and stroma thylakoids were severely linear, grana cannot be stacked normally, and the chloroplast ultrastructure was damaged in leaves of ‘ygl9’ at seeding stage”

25) Line 257 - 258. It is better to delete this sentence.

Response 25): Line 273 - 274. Delete this sentence.

26) Line 257 - 280. It is not clear how carotenoid metabolism pathway is related to color mutation.

Response 26): By regulating plant carotenoid synthase and plastid development (carotenoid metabolism pathway), the carotenoid content can be changed, which affects the ratio of pigments and the development of chloroplasts, resulting in leaf color mutations.

The following modifications have been made in this paragraph:

(1) Line 284 - 286. “…and a corresponding increase in total carotenoids” => “…and a corresponding increase in total carotenoids that facilitate greening when etiolated seedlings emerge from the soil”

(2) Line 286 – 288. Add “Ectopic expression of PSY1 in tobacco leads to leaves showed abnormal pigmentation, and very young leaves sometimes were colored bright orange, but then rapidly turned green [78].” after “…when etiolated seedlings emerge from the soil [77]”

References:

[78] Busch, M.; Seuter, A.; Hain, R. Functional analysis of the early steps of carotenoid biosynthesis in tobacco. Plant physiology 2002, 128, 439-453, doi:10.1104/pp.128.2.439.

(3) Line 288 - 295. “Increases in carotenoid content in orange and tomato plants have also been positively correlated with phytoene synthase gene expression [78,79]. When citrus fruits mature, the LCYB expression increases and LCYE disappears, while β-ring zeaxanthin and violaxanthin increase significantly [80]” => “The immutans (im) variegation mutant of Arabidopsis has light-dependent variegated phenotypes (green and white leaf sectors), the green leaf sectors contain normal chloroplasts, while the white leaf sectors contain abnormal chloroplasts that lack carotenoids due to a defect in phytoene desaturase activity [79,80].”       

References:

[79]  Aluru, M.R.; Bae, H.; Wu, D.; Rodermel, S.R. The Arabidopsis immutans mutation affects plastid differentiation and the morphogenesis of white and green sectors in variegated plants. Plant physiology 2001, 127, 67-77, doi:10.1104/pp.127.1.67.

[80] Aluru, M.R.; Zola, J.; Foudree, A.; Rodermel, S.R. Chloroplast Photooxidation-Induced Transcriptome Reprogramming in Arabidopsis immutans White Leaf Sectors. Plant Physiology 2009, 150, 904-923, doi:10.1104/pp.109.135780.

27) Line 355 - 356. This mutant may be of fruit, not of leaves.

Response 27):  Line 377 - 382. “Mutation of the anthocyanin-specific regulatory genes VvMYBA1 and VvMYBA2 resulted in the abnormal expression of anthocyanin-specific structural gene 3GT,which resulted in white grape mutants”   =>   “PtrMYB119 function as transcriptional activators of anthocyanin accumulation in both Arabidopsis and poplar. Overexpression of PtrMYB119 in hybrid poplar resulted in elevated accumulation of anthocyanins in whole plants, which had strong red-color pigmentation especially in leaves relative to non-transformed control plants”

References:

[101] Cho, J.-S.; Van Phap, N.; Jeon, H.-W.; Kim, M.-H.; Eom, S.H.; Lim, Y.J.; Kim, W.-C.; Park, E.-J.; Choi, Y.-I.; Ko, J.-H. Overexpression of PtrMYB119, a R2R3-MYB transcription factor from Populus trichocarpa, promotes anthocyanin production in hybrid poplar. Tree Physiology 2016, 36, 1162-1176, doi:10.1093/treephys/tpw046.

28) Line 365 - 369. These mutants may be of flowers, not of leaves.

Response 28):

(1) Line 392 – 400. “In octoploid dahlia (Dahlia pinnata), bHLH class regulatory genes were inserted into a transposon and inhibited anthocyanin synthesis, which resulted in the formation of the mottled phenotype of the yellow bottom orange spot [105]. Additionally, the ectopic expression of the bHLH genes NtAn1a and NtAn1b in tobacco flowers increased anthocyanin content, and the NtAn1-NtAn2 complex could activate promoters of CHS and DFR [106]”  =>  

“In Setaria italica, PPLS1 is a bHLH transcription factor that regulates the purple color of pulvinus and leaf sheath in foxtail millet. Co-expression of both PPLS1 and SiMYB85 in tobacco leaves increased anthocyanin accumulation and the expression of genes involved in anthocyanin biosynthesis (NtF3H, NtDFR and Nt3GT), resulting in purple leaves [105]”

References:

[105] Bai, H.; Song, Z.; Zhang, Y.; Li, Z.; Wang, Y.; Liu, X.; Ma, J.; Quan, J.; Wu, X.; Liu, M., et al. The bHLH transcription factor PPLS1 regulates the color of pulvinus and leaf sheath in foxtail millet (Setaria italica). Theoretical and Applied Genetics 2020, 133, 1911-1926, doi:10.1007/s00122-020-03566-4.

(2) Line 403-408. Add “The regulatory R2R3-MYB transcription factor AmRosea1 and the bHLH transcription factor AmDelila in Antirrhinum majus have been extensively studied. The simultaneous expression of the AmRosea1 and AmDelila activates anthocyanin production in the orange carrot, which young leaves were dark purple and matured into dark green leaves during the subsequent growth relative to non-transformed plants [107].” before “In addition, the chrysanthemum CmbHLH transcription…”

References:

[107] Sharma, S.; Holme, I.B.; Dionisio, G.; Kodama, M.; Dzhanfezova, T.; Joernsgaard, B.; Brinch-Pedersen, H. Cyanidin based anthocyanin biosynthesis in orange carrot is restored by expression ofAmRosea1andAmDelila,MYB and bHLH transcription factors. Plant Molecular Biology 2020, 103, 443-456, doi:10.1007/s11103-020-01002-1.

29) Line 427. Please add the results for carotenoid and anthocyanin reported in [122].

Response 29):

Line 471 – 483. Add the results for carotenoid and anthocyanin reported in [122](now is [127]), as is follow:

“For example, after RNA sequencing of wild type and rubescent mutants of Anthurium andraeanum ‘Sonate’, most gene expression levels in the mutants were downregulated compared with the wild type, which was related to chlorophyll synthesis, chloroplast development and differentiation” =>

“For example, in the mutants (etiolated, rubescent, and albino) of Anthurium andraeanum ‘Sonate’, the ultrastructure of chloroplasts in mutant leaves were disrupted, very few intact chloroplasts were observed. The ratio of carotenoid/total Chl in all 3 mutants was higher than that of the wild type, the content of anthocyanin was at a similar level in the leaf of etiolated mutant and wild type plants while the albino leaves had the lowest anthocyanin content. After RNA sequencing, most genes related to chlorophyll synthesis, anthocyanin transport and pigment biosynthesis were down-regulated compared with the wild type. These results indicate the abnormal leaf color of mutants was caused by changes in the expression pattern of genes responsible for pigment biosynthesis”

Reviewer 2 Report

Authors covered most of the aspects of pigment biosynthesis and mutation studies related to leaf color. This review looks good to me but I suggest authors to add one para on "how pigment or leaf color studies are important in current scenario of big data, genome and pan genome era." I think, this will open and avenue to young researchers working in pigment biology to explore the exiting genomic data and get some new information. In addition, I suggest authors to break the very last sentence (line 446-451) of "conclusion" section into two or more sentences as it looks very long to me. I wish good luck to authors.

Author Response

1)This review looks good to me but I suggest authors to add one para on "how pigment or leaf color studies are important in current scenario of big data, genome and pan genome era."

Response 1): Add one para:

“With the rapid development of next-generation genome sequencing technologies, the application of genome, pangenome and transcriptome is extensively increasing in revealing the genetic variation traits and dissecting the genetic regulation mechanism of key genes in plants. Current research is focused on revealing the mechanisms of growth, fruit quality, stress responses, mutation traits as well as gender determination and flowering in plants [113 -117]. Currently, map-based cloning and RNA-seq are mainly used to mine genes related to leaf color mutations. By using next-generation genome sequencing technologies to analysis genomic and transcriptomic data, can greatly improve the efficiency and precision of identification of leaf color mutation related genes in plant. The identification, cloning and functional study of these mutant genes play important roles in elucidating the molecular mechanisms of leaf color mutation, plant photosynthesis, pigment synthesis, etc.

Line 424 – 439.

“With the rapid development of genomics and molecular biology, research on the location and cloning of leaf color mutation related genes has become a hot topic both at home and abroad. The cloning and functional study of these mutant genes play important roles in elucidating the molecular mechanisms of leaf color mutation, plant photosynthesis, chlorophyll biosynthesis, etc. Currently, map-based cloning and RNA-seq are mainly used to mine genes related to leaf color mutations.”       =>

“With the rapid development of next-generation genome sequencing technologies, the application of genome, pangenome and transcriptome is extensively increasing in revealing the genetic variation traits and dissecting the genetic regulation mechanism of key genes in plants. Current research is focused on revealing the mechanisms of growth, fruit quality, stress responses, mutation traits as well as gender determination and flowering in plants [113 -117]. Currently, map-based cloning and RNA-seq are mainly used to mine genes related to leaf color mutations. By using next-generation genome sequencing technologies to analysis genomic and transcriptomic data, can greatly improve the efficiency and precision of identification of leaf color mutation related genes in plant. The identification, cloning and functional study of these mutant genes play important roles in elucidating the molecular mechanisms of leaf color mutation, plant photosynthesis, pigment synthesis, etc.”

References:

[113] Neale, D.B.; Kremer, A. Forest tree genomics: growing resources and applications. Nature Reviews Genetics 2011, 12, 111-122, doi:10.1038/nrg2931.

[114] Chen, S.; Lin, X.; Zhang, D.; Li, Q.; Zhao, X.; Chen, S. Genome-Wide Analysis of NAC Gene Family in Betula pendula. Forests 2019, 10, doi:10.3390/f10090741.

[115] Wang, S.; Huang, H.; Han, R.; Liu, C.; Qiu, Z.; Liu, G.; Chen, S.; Jiang, J. Negative feedback loop between BpAP1 and BpPI/BpDEF heterodimer in Betula platyphylla x B. pendula. Plant Science 2019, 289, doi:10.1016/j.plantsci.2019.110280.

[116] Voichek, Y.; Weigel, D. Identifying genetic variants underlying phenotypic variation in plants without complete genomes. Nature Genetics 2020, 52, 534-+, doi:10.1038/s41588-020-0612-7.

[117] Liu, Y.; Du, H.; Li, P.; Shen, Y.; Peng, H.; Liu, S.; Zhou, G.-A.; Zhang, H.; Liu, Z.; Shi, M., et al. Pan-Genome of Wild and Cultivated Soybeans. Cell 2020, 182, 162-+, doi:10.1016/j.cell.2020.05.023.

2)Line. 446 – 451. I suggest authors to break the very last sentence (line 446-451) of "conclusion" section into two or more sentences as it looks very long to me. 

Response 2): Break this sentence.

Line. 502 - 512.

“Therefore, in future studies, we should not only strengthen the research on other pigment molecules, nuclear-plasmid signal transduction, transcription regulation, etc., but we should also examine noncoding RNA, DNA methylation and so on, which will help us to obtain more relevant information on the regulation of pigment anabolic metabolism and to analyze the molecular regulation mechanisms of plant leaf color mutations from a system network level”      =>      “Therefore, in future studies, we should strengthen the research on other pigment molecules, nuclear-plasmid signal transduction, transcription regulation as well as examine noncoding RNA, DNA methylation, etc. This will help us to obtain more relevant information on the regulation of pigment anabolic metabolism, and to analyze the molecular regulation mechanisms of plant leaf color mutations from a system network level”
